# Safety and tolerability of tedizolid as oral treatment for bone and joint infections

Loren G. Miller,[1,2] Evelyn A. Flores,[1] Bryn Launer,[1] Pamela Lee,[1] Praneet Kalkat,[1] Kelli Derrah,[3] Shalini Agrawal,[1] Matthew Schwartz,[1] Grant Steele,[4] Tae Kim,[1] Maita S. Kuvhenguhwa[5]

**ABSTRACT** Bone and joint infections (BJIs) are common infections increasingly managed with oral therapy. However, there are limited safe oral options for many Gram-positive pathogens. In animal studies and short-term human use, tedizolid lacks the hematologic and neurologic toxicity of the other available oxazolidinone, linezolid. However, there are limited prospective safety data. We conducted an open-label, non-comparative trial of oral tedizolid for BJI treatment. Primary outcomes were safety and cure rate. Eligible patients had a BJI caused by documented or suspected Gram-positive pathogen, required 4–12 weeks of therapy, and did not have myelosuppression or peripheral/optic neuropathy. Subjects underwent weekly evaluation for cytopenias and neuropathy. We enrolled 44 subjects; five were lost to follow-up. Two subjects did not complete planned treatment because of rash ($n = 1$) and urgent surgery ($n = 1$). Of 37 patients with evaluable outcomes, 17 (46%) had hardware-associated infection, 13 (35%) had osteomyelitis, 5 (14%) had prosthetic joint infection, and 2 (5%) had other BJIs. Median (mean, range) treatment duration was 12 (10.1, 4–12) weeks. There were no cases of cytopenias or peripheral or optic neuropathy. Treatment cure occurred in 13 (35%); 19 (51%) required antibiotic continuation after 12 weeks of tedizolid related to retained hardware at the BJI site, and failure occurred in four (11%), two unlikely, one possibly, and one probably due to tedizolid. We found that oral tedizolid was well tolerated for prolonged BJI treatment without significant toxicity. Clinical failure rate was similar to that of other published BJI investigations. (This study has been registered at Clinicaltrials.gov under identifier NCT03009045.)

**IMPORTANCE** Bone and joint infections are common infections with limited effective and safe oral options for Gram-positive infections. The largest prospective clinical trial of tedizolid therapy for bone and joint infections enrolled 44 patients and tested each in person weekly with detailed safety monitoring including tests for leukopenia, anemia, thrombocytopenia, peripheral neuropathy, and optic neuropathy for up to 12 weeks. Findings demonstrated tedizolid was generally well tolerated and there were no incident cases of cytopenias or neuropathy. Cure rates were similar to that in other bone and joint infection studies. In summary, oral tedizolid appears to be a well-tolerated oral option for Gram-positive bone and joint infections.

**KEYWORDS** tedizolid, bone and joint infection, Gram-positive bacteria

Bone and joint infections (BJIs) are common infections that remain challenging to treat. Recent data suggest that the incidence of BJIs are increasing dramatically (1–3). Population-based U.S. data from Olmsted County showed that the incidence of osteomyelitis has doubled from rates 30 years prior (3). BJI treatment is particularly problematic because it requires prolonged therapy for weeks or months, which incurs risk of treatment-related and treatment-limiting side effects (4).

Address correspondence to Loren G. Miller, Lgmiller@ucla.edu.

This project was supported by a grant from Merck to Loren Miller. The sponsor had no role in study design, conduct, interpretation, or manuscript preparation. Loren Miller has received grants from GSK, Paratek, ContraFect, and Medline Industries.

See the funding table on p. 9.

BJI treatment is not standardized, which allows a wide range of potential therapies. Oral antibiotics are increasingly accepted, as a recent randomized clinical trial found that oral antibiotics for BJI achieved similar cure rates to intravenous therapies (5). Oral therapies also avoid cost and risk associated with intravenous therapy (4).

Gram-positive pathogens are the dominant causes of BJIs. *Staphylococcus aureus* is the most common cause of osteomyelitis (3). Methicillin-resistant *S. aureus* (MRSA) are increasingly common and roughly half of all *S. aureus* are MRSA in most U.S. and non-international centers (6). Many *S. aureus* and MRSA isolates have limited oral antibiotic options (6) and these available antibiotics have limitations. Clindamycin resistance is increasing with rates as high as 40% among MRSA (7). Trimethoprim-sulfamethoxazole has been associated with acute kidney injury and hyperkalemia (8, 9). Linezolid, an oxazolidinone, is associated with serious toxicities such as cytopenias, peripheral neuropathy, and optic neuropathy with prolonged therapy (10).

Tedizolid, a newer oxazolidinone antibiotic, has a favorable adverse event profile and low thrombocytopenia rates compared to linezolid (11, 12). Drug-mitochondrial binding, which is believed to be the mechanism by which mid- and long-term toxicities of linezolid such as neuropathy are acquired, is not seen in animal models and human data with tedizolid, suggesting that the side effects associated with long-term therapy of linezolid may not be found with tedizolid (12).

Given tedizolid's promising profile for prolonged administration, we conducted an open-label trial of oral tedizolid for BJI treatment.

## MATERIALS AND METHODS

### Study population

Participants were recruited from outpatient and inpatient sites at Harbor-UCLA Medical Center, Torrance CA. We also provided a flyer to an infectious diseases physician group at a nearby medical center to refer patients to our center for possible participation, as clinically indicated. Inclusion criteria included BJI caused by a documented or suspected Gram-positive organism, age ≥18, and plans for outpatient BJI treatment. Exclusion criteria included plans for hospitalization >1 week, pregnancy, uncontrolled comorbidities (e.g., diabetes, psychiatric disease), planned indefinite treatment course (e.g., chronic suppressive antibiotics for life), peripheral or optic neuropathy, underlying hematologic cytopenias, severe hepatic dysfunction, hypersensitivity to tedizolid or other oxazolidinones, ongoing antibiotic-associated colitis, a diet high in tyramine-containing foods, concurrent use of drugs that may interact with tedizolid (specifically sodium picosulfate or monoamine oxidase inhibitors), previous study participation, and use of tedizolid for any condition in the prior 3 months.

### Treatment

All participants were given tedizolid 200 mg once daily for their documented or suspected Gram-positive infection. If participant's providers noted documented or suspected infection caused by other non-Gram-positive pathogens, co-administration of other antibiotics was allowed. Treatment with tedizolid was limited to 12 weeks. If patients required continued antibiotic therapy after this duration, then treatment would be continued by the treating physician with an antibiotic of that physician's choices.

### Data collection

On enrollment, we collected participants' demographics, comorbidities, surgical history, pertinent microbiology results, and concomitant medications using standardized forms. At baseline, we also collected complete blood counts (CBC) and comprehensive metabolic panels (CMP), which included electrolytes and liver function tests. Susceptibility testing to tedizolid or linezolid was not routinely performed given that susceptibility among *S. aureus* and Gram-positive pathogens to these antibiotics is universal (13) and

many isolates were no longer obtainable from the Clinical Microbiology Laboratory when participants consented to participation. Monofilament testing was done on all subjects to assess for previously undetected peripheral neuropathy. We gave each participant a medication diary and asked them to record the time of day when they took each pill as well as any side effects or comments about the medication or their symptoms. Participants were prescribed tedizolid by a research pharmacy and provided bottles that contained 30-day supplies.

After enrollment, participants came to a research clinic weekly where we performed CBCs and CMP weekly. Additionally, we administered standardized surveys for drug side effects and tolerability, adverse events, unplanned surgical procedures, and signs or symptoms of possible recurrent BJI. Participants were asked to bring their study drug medication bottle and completed medication diary. Participants completed weekly visits for the duration of their tedizolid treatment (up to 12 weeks). Participants also came for a visit 1 week post-tedizolid treatment completion where the same procedures as outlined above were completed. Finally, we conducted a phone survey 3 months post-tedizolid treatment completion. This standardized survey asked participants about their overall well-being, recurrence of signs or symptoms of BJI, and any self-reported medical conditions or adverse events.

## Study outcomes

Our primary outcomes were safety and efficacy. Safety was defined as the number of "moderate" and "serious" adverse events using MedDRA 18.1 definitions. Severity of adverse events was measured using the CTCAE 5.0 system (14). Safety was monitored via weekly in-person examinations and blood monitoring, as outlined above. We also measured safety outcomes that were "mild and defined cytopenias as leukocytes $<4.0 \times 10^9$ /L, hemoglobin <8.0 g/dL, and platelets <150,000/mcL.

BJI treatment outcome categories were defined *a priori* and were resolution, failure, including its relation to the study drug (related, probably relatedly, possibly related, unlikely related), and ongoing therapy required. For the latter category, study physicians categorized whether the ongoing therapy could have been related to suboptimal study drug efficacy (likely, probably, possibly, unlikely). All infection outcomes (efficacy) were determined independently by two infectious disease physicians who were not part of the investigative team through review of medical and study record using the above definitions. If there was disagreement between the physicians that could not be resolved with discussion, a third study physician arbitrated for the final determination.

## Statistical analysis

Study results were descriptive. We calculated means of study laboratory results including leukocytes, hemoglobin, platelets, aspartate aminotransferase (AST), alanine transaminase (ALT), and alkaline phosphatase (ALP). Secular trends in laboratory test result trends were assessed using the Mann-Kendall trend test.

## RESULTS

We screened 570 patients with BJI diagnosis. We excluded 526 patients; common reasons for exclusion included not meeting inclusion criteria ($N = 331$), peripheral and/or optic neuropathy ($n = 55$), treating physician not agreeable to tedizolid treatment ($n = 39$), investigator concerns for patient not completing study participation ($n = 32$), no-shows to clinic visit ($n = 29$), underlying hematologic abnormalities ($n = 22$), and uncontrolled comorbidities ($n = 7$). Other reasons are outlined in Fig. 1.

We enrolled 44 (7.7%) subjects. None had previously undetected peripheral neuropathy by monofilament testing. Of these, we excluded five from evaluation because of loss to follow-up relatively soon after enrollment ($n = 3$), provider changed treatment plan ($n = 1$), and subject moved >100 miles away and could not attend follow-up visits ($n = 1$). Two additional subjects received incomplete courses of tedizolid;

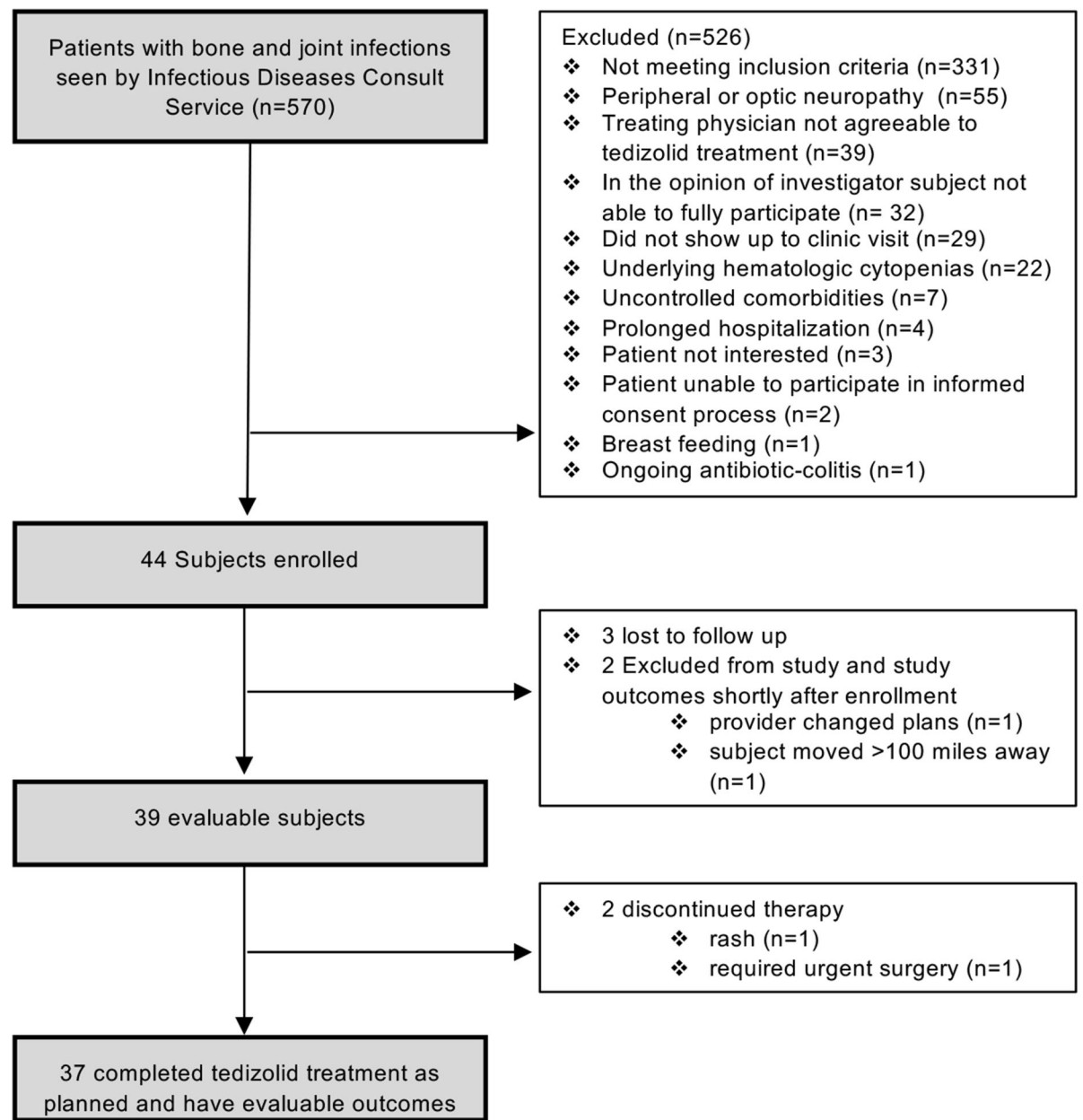

**FIG 1** The results detail the screening process to identify patients eligible for the clinical trial and reasons for exclusion and non-participation. The results also detail reasons for patients leaving the study.

one withdrew due to tedizolid-associated rash after 4 weeks of therapy (Subject #3, Table S1), and one withdrew from the study after undergoing urgent surgery 2 weeks after tedizolid initiation (Subject #20). All subjects' clinical courses are summarized in Table S1.

Among the remaining 37 subjects with evaluable outcomes, mean (median, range, interquartile range) of tedizolid treatment duration was 10.1 (12, 4–12, 8–12) weeks. Baseline comorbidities included diabetes (without peripheral neuropathy) ($n = 11$, 30%), anxiety/depression ($n = 4$, 11%), systemic lupus ($n = 2$, 5%), asthma ($n = 2$, 5%), and substance abuse ($n = 2$, 5%); a full list is in Table 1. In terms of BJI type, 17 (46%) had hardware-associated infections. Thirteen (35%) had osteomyelitis that was not associated with hardware, including 5 (14%) that had hardware removed recently (Table 1). Other anatomical sites are outlined in Table 1. Other infections included prosthetic joint infection (post-intravenous therapy, $n = 5$, 14%), external fixator-associated infection

**TABLE 1** Demographics and clinical characteristics

| Variable | All (n = 37) |
|---|---|
| Age | |
| Mean | 44 |
| Median, mean (range) | 46, 45.0 (22–65) |
| Gender | |
| Female | 2 (5%) |
| Male | 35 (95%) |
| Race/Ethnicity | n (%) |
| Hispanic | 23 (62%) |
| White | 2 (5%) |
| Black/African American | 10 (27%) |
| Native American/Native Indian | 1 (3%) |
| Declined to state | 1 (3%) |
| Major comorbidities[a] | n (%) |
| Diabetes | 11 (30%) |
| Anxiety or depression | 4 (11%) |
| Systemic lupus erythematosus | 2 (5%) |
| Asthma | 2 (5%) |
| Substance abuse | 2 (5%) |
| Immune thrombocytopenic purpura | 1 (3%) |
| Rheumatoid arthritis | 1 (3%) |
| End-stage renal disease with hemodialysis | 1 (3%) |
| Chronic myelogenous leukemia | 1 (3%) |
| Crohn's disease | 1 (3%) |
| Immune thrombocytopenic purpura | 1 (3%) |
| Parapalegia | 1 (3%) |
| Seizure disorder | 1 (3%) |
| Type of bone or joint infection | n (%) |
| Hardware-associated infection | 17 (46%) |
| Osteomyelitis, non-hardware associated | 13 (35%) |
| Acromioclavicular | 1 (3%) |
| Calcaneal | 1 (3%) |
| Olecranon (with bursitis) | 1 (3%) |
| Post-hardware removal | 5 (14%) |
| Sacral | 1 (3%) |
| Shoulder | 1 (3%) |
| Spinal | 1 (3%) |
| Spinal with epidural abscess | 1 (3%) |
| Sternoclavicular | 1 (3%) |
| Prosthetic joint infection | 5 (14%) |
| External-fixator associated | 1 (3%) |
| Septic arthritis | 1 (3%) |
| Gram-positive causative pathogens[a] | n (%) |
| S. aureus | 17 (45%) |
| MSSA[b] | 13 (34%) |
| MRSA[c] | 4 (11%) |
| Coagulase-negative Staphylococcus | 6 (16%) |
| Enterococcus faecalis | 4 (11%) |
| Streptococcus agalactiae | 2 (5%) |
| Cutibacterium acnes | 2 (5%) |
| Enterococcus faecium, vancomycin resistant | 1 (3%) |
| Viridans group Streptococcus | 1 (3%) |

(*Continued on next page*)

**TABLE 1** Demographics and clinical characteristics (*Continued*)

| Variable | All (*n* = 37) |
|---|---|
| No Gram-positive pathogen identified, but suspected Gram-positive etiology | 6 (16%) |
| Concomitant non-Gram-positive pathogen | 7 (18%) |

[a]Totals do not add up to 100% as some subjects had no significant comorbidities and some subjects had >1 Gram-positive pathogen (see Table S1).
[b]MSSA, methicillin-susceptible *Staphylococcus aureus*.
[c]MRSA, methicillin-resistant *Staphylococcus aureus*.

(*n* = 1, 3%), and septic arthritis (*n* = 1, 3%). Pathogens associated with infection included *Staphylococcus aureus* (*n* = 17, 45%), coagulase negative *Staphylococcus* (*n* = 6, 16%), *Enterococcus faecalis* (*n* = 4, 11%), *Streptococcus agalactiae* (*n* = 2, 5%), *Cutibacterium acnes* (*n* = 2, 5%), vancomycin-resistant *Enterococcus faecium* (*n* = 1, 3%), and viridans group *Streptococcus* (*n* = 1, 3%). Six subjects (16%) had no Gram-positive pathogen identified but had suspected Gram-positive infection. Seven subjects (18%) had a concomitant non-Gram-positive pathogen and received concomitant antibiotics for these organisms (Table S1). Twenty-five (68%) had received prior antibiotic therapy for their BJI; no patient had received prior linezolid therapy (Table S1).

## Safety monitoring

There were zero moderate or serious adverse events. There were no incident episodes of leukopenia, thrombocytopenia, or anemia. Additionally, there were no incident episodes of abnormalities of ALT, AST, or ALP. At baseline, mean (standard deviation) leukocyte count was $7.04 \times 10^9$/L (2.22, *n* = 38). At Week 12, mean leukocyte count was $6.95 \times 10^9$/L (2.98, *n* = 28), which was not a significant change ($P = 0.47$) (Fig. S1). Mean (standard deviation) hemoglobin at baseline was 12.66g/dL (2.01) and at Week 12 was 13.59 (1.76 g/dL, *n* = 28). This was a significant change ($P = 0.005$) with a trend of increased hemoglobin over the course of monitoring (Kendall tau-b correlation coefficient = 0.56). Platelets at baseline had a mean (standard deviation) at baseline of 313,000/mcL (117,100) and changed to 297,250/mcL (93,770, *n* = 28), which was not a significant change ($P = 0.70$). There were no significant changes in AST, ALT, or ALP (Fig. S1) ($P > 0.05$ for all comparisons). There were also no new electrolyte abnormalities seen (data not shown).

During weekly assessment of peripheral or optic neuropathy, no subjects developed new symptoms during or after the treatment course. One subject developed new "light bursts" after 8 weeks of tedizolid. Subject was promptly seen by an ophthalmologist with retinal specialty who found no ocular or retinal pathology. Symptoms then resolved spontaneously.

## Efficacy

Of the 37 subjects who completed their planned tedizolid course, 13 (35%) achieved clinical cure. Nineteen (51%) subjects required ongoing therapy at the end of their planned treatment course, typically because of ongoing infection associated with retained hardware that could not be removed after Week 12 of tedizolid treatment. In the opinion of the treating clinicians (with confirmation by the research physicians), 18 of 19 subjects (95%) requiring ongoing therapy after the planned course of tedizolid were unlikely due to tedizolid failure and 1 (5%) was possibly due to tedizolid failure (Subject #6, Table S1). At the 3-month follow-up call, 17 (46%) described continued improvement or clinical stability and 20 (54%) did not respond to the survey. No patient described worsening BJI or relapse at 3-month follow-up.

Four (11%) of the subjects had clinical failure. Of these, one (3%) was felt to be probably due to the study drug, one (3%) was possibly due to the study drug (Subject #7, Table S1), and two (5%) were felt to be unrelated to the study drug. None of the failures had culturable material; thus, we could not test for oxazolidinone-resistant pathogens.

One subject (Subject #37) was felt to have indeterminate outcome as he had improvement in symptoms of septic arthritis with tedizolid, but 10 days later, the subject had worsening of joint pain and was found to have purulent discharge in a fluid space not contiguous with the original infection. Details on individual cases are outlined in Table S1.

## DISCUSSION

BJIs are an increasingly common problem in clinical care (1–3). As antimicrobial resistance rises among Gram-positive organisms such as *S. aureus*, there is an increasing need for safe and effective options for BJI treatment. In our open-label study focusing on tedizolid safety in BJI treatment, the largest and most systematic investigation on tedizolid safety to date, we found that that prolonged treatment with tedizolid (mean = 10.1 weeks, median = 12 weeks) was well tolerated with no detected hematologic or neurologic side effects.

Tedizolid is mechanistically distinct from the other currently available oxazolidinone, linezolid. The lower incidence of thrombocytopenia with tedizolid compared to linezolid (15) is likely due to its differential potential to inhibit mitochondrial protein synthesis, as demonstrated in pre-clinical studies (16). At FDA-approved doses, tedizolid has low rates of hematologic side effects (11, 12). Additionally, unlike linezolid, interactions with selective serotonin reuptake inhibitors appear unlikely (12). Mitochondrial toxicity from linezolid can drive cytopenias and neuropathy (17). These adverse events were rarely seen with tedizolid, suggesting that the side effects associated with long-term therapy of linezolid may not be found with tedizolid (12). Our findings support the safety of prolonged therapy with tedizolid.

Our study's primary outcomes were safety and cure rate. Among the 38 subjects who received at least 4 weeks of therapy, the sole discontinuation occurred due to a non-life-threatening maculopapular rash. When the rash occurred at treatment Week 3, the subject and clinicians initially felt the rash was mild. After 2 additional weeks of spreading rash, the subject felt the rash was bothersome enough for discontinuation. The rash then resolved. We found no cases of new anemia, leukopenia, or thrombocytopenia among study subjects. Additionally, we saw no significant decreases in leukocyte count, or thrombocyte counts over time. We did find a significant increase in hemoglobin over the course of treatment. It should be noted that the number of participants who underwent laboratory testing decreased over time so the number under study was not constant. Finally, we systematically surveyed all subjects weekly for new symptoms of peripheral or optic neuropathy and no patients developed new clinically identifiable neuropathy.

Our findings are similar to that of Morisette et al., who retrospectively looked at safety of prolonged tedizolid in 37 subjects who took tedizolid for a mean of 188 days (18), compared to our median of 96 days. In their study, they similarly found no new clinically significant hematologic cytopenias or neuropathies. They did have three patients who required tedizolid discontinuation due to (one episode each of) macrocytic anemia, lactic acidosis, and dizziness, although in the latter two cases, these symptoms were felt to be likely attributable to other medications. Additionally, their investigation differed in that most (65%) patients were given antibiotics for suppression of chronic infection.

Ferry et al. prospectively looked at the safety of suppressive treatment in implant-associated BJIs in 17 patients with multidrug-resistant Gram-positive infections and found tedizolid to be well tolerated (19). Senneville et al. prospectively examined 33 patients with prolonged therapy undergoing surgical intervention for a mean of 8.0 weeks (20). Two patients (6%) had to stop tedizolid due to intolerance, and two more (6%) due to anemia, although the anemia was felt to be due to bleeding. York et al. retrospectively examined 62 patients who received a mean of 27 days of tedizolid (21). Most patients had BJI and had stopped linezolid because of toxicity; they found that hematologic adverse events were infrequent. Similarly, Mensa Vendrell et al. retrospectively examined 81 patients on prolonged tedizolid (median duration 26.5 days), many of

whom experienced linezolid toxicity; among these 81 patients, 38 had BJIs (22). Although tedizolid was generally well tolerated, they found a 1% incidence of anemia and a 7% incidence of thrombocytopenia. Finally, Benavet et al. retrospectively examined 51 cases of osteoarticular infections, mostly (92%) as part of an antibiotic therapy switch (23). Like the other investigations, they found that tedizolid was well tolerated with three patients (6%) who had nausea and occasional vomiting, none of which required tedizolid discontinuation.

Our study differed from some of the above studies as we used tedizolid for treatment of infection with the intention of clinical cure rather than suppression. Additionally, unlike all of the above studies, we systematically surveyed for cytopenias, electrolyte and liver function abnormalities, and neuropathies weekly.

In terms of efficacy, we found a treatment failure rate of 11%, similar to the rates of 6%–24% reported by others (18–20, 23). How these treatment failure rates compare to that of other antimicrobial regimens, intravenous or oral, is unclear. BJI cure rates vary widely by clinical indication and the population under study (1, 2, 4, 5). Thus, benchmarking tedizolid cure rates to that of other antibiotics outside of a randomized controlled trial would be highly problematic. Nevertheless, the failure rates seen with tedizolid treatment by our study and others suggest a comparable efficacy for BJIs. Finally, while our cure rate was only 35%, we utilized a third outcome category, "ongoing therapy required," which occurred in 51% of study participants. This outcome was used given the nature of the disease under study—BJIs with retained hardware that could not be removed until adequate bone healing. While these "need for ongoing therapy" outcomes were not cures *per se*, they also did not represent failure and instead represented the requirement of additional antibiotics for complex BJIs that often typically required source control, i.e., hardware removal.

There are limitations to our study. First, our study was of limited by sample size (37 subjects who completed planned tedizolid therapy). However, the consistency of our findings with other investigations is reassuring that the safety and efficacy we found was likely representative of patients with BJIs that are treated with tedizolid. Second, our study was non-comparative. While we found tedizolid to be safe, its relative safety compared to other regimens is not defined in our investigation. Regardless, the tolerability of tedizolid in this study and others strongly suggest that tedizolid is generally well tolerated and side effects causing treatment discontinuations are uncommon. Third, we excluded patients with hematologic cytopenias, severe hepatic dysfunction, and use of drugs that interact with tedizolid, thus limiting the generalizability of findings to broader populations. Finally, our follow-up was limited and many subjects did not complete the phone survey 3 months after treatment completion. BJI relapse can occur weeks or months after treatment discontinuation (4), so the true treatment failure rate of tedizolid in our cohort may have been underestimated. However, among those who completed the 3-month post-treatment survey, none had relapse.

There are strengths to our study. As noted above, ours is the largest prospective investigation of tedizolid treatment for BJIs. Second, we used systematic measures of surveying for laboratory abnormalities as well as new neurologic symptoms. Finally, our subjects had a diversity of BJI types, including native bone infections that were not studied (19, 20) or formed only a minority of patients (18, 23) in the other investigations of tedizolid therapy.

In summary, under close and systematic evaluation, we found tedizolid to be a well-tolerated oral therapy for BJIs with no significant associated hematologic or neurologic side effects in our cohort. Based on our findings, tedizolid should be considered an option for BJI treatment and may have an important role as oral option for difficult-to-treat multidrug-resistant Gram-positive organisms.

## ACKNOWLEDGMENTS

We appreciate the assistance of The Lundquist Institute CTRU nursing staff, Dina Wilson, Eritrea Keleta, and Loritta Manai, Roxanne Tanoviceanu, and Hannah Mansky of the Lundquist Institute Investigational Drug Services. We also appreciate the assistance of Deborah Kupferwasser, DrPH, and Gregory Tchakalian for their help with data analysis and management.

We especially appreciate our study participants for their participation in this trial.

This study was supported by a grant from Merck to L.G.M. The sponsor had no role in study design, study conduct, data analysis, or result interpretation.

L.G.M. has received grant funding from Merck. All other authors have no conflicts of interest to disclose.

## AUTHOR AFFILIATIONS

[1]Lundquist Institute for Biomedical Research at Harbor-UCLA Medical Center, Torrance, California, USA
[2]Lundquist David Geffen School of Medicine at UCLA, Los Angeles, California, USA
[3]University of California Davis Medical Center, Sacramento, California, USA
[4]Harvard Medical School, Boston, Massachusetts, USA
[5]MLK Community Healthcare, Los Angeles, California, USA

## AUTHOR ORCIDs

Loren G. Miller  http://orcid.org/0000-0003-0487-1711

## FUNDING

| Funder | Grant(s) | Author(s) |
| --- | --- | --- |
| Merck (Merck & Co.) | 021814-01-00 | Loren G. Miller |

## ETHICS APPROVAL

Written informed consent was obtained from the participant in their preferred language (English or Spanish). The protocol was approved by the Institutional Review Board of The Lundquist Institute at Harbor-UCLA Medical Center.

## ADDITIONAL FILES

The following material is available online.

### Supplemental Material

**Supplemental Material (Spectrum01282-23-s0001.pdf).** Table S1 and Figure S1.

### Open Peer Review

**PEER REVIEW HISTORY (review-history.pdf).** An accounting of the reviewer comments and feedback.

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
