## [Reviewer comments · Microbiology Spectrum]

Microbiology Spectrum

Safety and Tolerability of Tedizolid as Oral Treatment for Bone and Joint Infections

Loren Miller, Evelyn Flores, Bryn Launer, Pamela Lee, Praneet Kalkat, Kelli Derrah, Shalini Agrawal, Matthew Schwartz, Grant Steele, Tae Kim, and Maita Kuvhengahwa

Corresponding Author(s): Loren Miller, The Lundquist Institute

Review Timeline:

Submission Date:	April 3, 2023
Editorial Decision:	April 23, 2023
Revision Received:	July 27, 2023
Accepted:	August 7, 2023

Editor: Adriana Rosato

Reviewer(s): Disclosure of reviewer identity is with reference to reviewer comments included in decision letter(s). The following individuals involved in review of your submission have agreed to reveal their identity: Eric Senneville (Reviewer #2)

Transaction Report:

DOI: <https://doi.org/10.1128/spectrum.01282-23>

April 23, 2023

Dr. Loren G. Miller
The Lundquist Institute
Infectious Diseases
Harbor-UCLA Medical Center
1000 W. Carson St., Box 466
Torrance, CA 90509

Re: Spectrum01282-23 (Safety and Tolerability of Tedizolid as Oral Treatment for Bone and Joint Infections)

Dear Dr. Loren G. Miller:

Link Not Available

Sincerely,

Adriana Rosato

Journals Department
Reviewer comments:

Reviewer #1 (Comments for the Author):

As prior reviewer #2, I am paying particular attention to the revisions they have made in response to my earlier comments. Overall, they have done a mostly acceptable job addressing these comments. They have brought the study outcomes closer in line with the clinicaltrials.gov site which was my major concern in this unblinded study - specifically that they may not be following their own pre-specified analysis plan. I would still note that the primary outcome measure listed on the clinicaltrials.gov is not reflected in how the manuscript is presented. Pasted here for completeness, the co-primary outcome was defined as:

1) Number of "moderate" or "serious" adverse events as assessed by MedDRA v18.1 [Time Frame: 4-12 Weeks]

2) Number of participants with an outcome of "cure" as defined as no need for further antibiotics beyond the originally planned duration determined by the participant's primary/treating physician. [Time Frame: 4-12 Weeks]

Given this endpoint, I would still expect to see these two measures presented, probably in that order, in the abstract and the manuscript. As in, "x number of solicited moderate or severe adverse events were observed in patients on tedizolid." However the abstract does not provide this. Also that endpoint is not entirely clear since MedDRA is not a tool for grading severity, it's for classifying events into standardized terminology - how was severity assessed?

Minor comments:

I would respectfully submit that despite the claim of careful re-review, they still have not caught or corrected the typo in the abstract results section where it says "...and 5 had (14%) prosthetic joint infection." It should not read "5 had (14%)"

Another typo: sentence fragment in the study outcomes section - "Adverse events using MedDRA 18.1 definitions."

Reviewer #2 (Comments for the Author):

The authors report the results of an open label, non-comparative trial of oral tedizolid for bone and joint infections. A total of 37 patients could be evaluated regarding tolerance and efficacy. It was found that prolonged treatment with oral tedizolid was well tolerated and that clinical failure rate was similar to that of other published reports.

General comments

The study is recorded in Clinicaltrials.gov

General comments

The subject is of interest given the paucity of data about the efficacy of tedizolid for the treatment of BJIs.

My main critics are about (i) the mix of very different types of BJIs about effectiveness and (ii) the very short post-treatment follow-up which both reduce the validity of the conclusions.

Detailed comments

1. BJI is a very vague term that included a lot of different situations associated with distinct outcomes.

2. The study was conducted paper with funding from Merk

3. Gram-positive pathogens : according to Cochrane = gram-positive (https://community.cochrane.org/book_pdf/295)

4. Line 100 : "uncontrolled comorbidities (e.g., diabetes, psychiatric disease)" and "underlying hematologic cytopenias, severe hepatic dysfunction, hypersensitivity to tedizolid or other oxazolidinones, ongoing antibiotic-associated colitis, a diet high in tyramine-containing foods, concurrent use of drugs that may interact with tedizolid (specifically sodium picosulfate or monoamine oxidase inhibitors)," Using all these restriction in the recruitment of the patients decreases the validity about the conclusion on the good profile of tolerability of Tedizolid compared to that of Linezolid as these restrictions correspond to some of the risks factors for Linezolid-related serious adverse events. It also means that the conclusions drawn from the present study only apply to these selected patients.

5. Data about the duration of antibiotics before Tedizolid was started are not presented in the manuscript.

6. Line 140: "Finally, we conducted a phone survey 3 months post-tedizolid treatment completion." This is a very short post-treatment follow-up given the nature of the diseases which is likely to have overestimated the remission rate recorded in the study.

7. What was the microbiology identified in the failure patients (any emergence of linezolid-resistant mutants)?

8. Line 249: "The lower incidence of thrombocytopenia with tedizolid compared to linezolid » What are the clinical data that support this assertion?

9. Line 304 : "Finally, while our cure rate was only 35%, ..." "remission" is a more accurate term than "cure" in the settings of BJIs especially in case of chronic infections.

10. The proportion of patients with "ongoing therapy required" is particularly high (51%); the reasons why this prolonged treatment was decided by the physicians need to be described in the manuscript in order to help make a distinction between palliative (stabilization of a failure) and suppressive treatment (maintain of a prolonged remissions).

Staff Comments:

Preparing Revision Guidelines

Please return the manuscript within 60 days; if you cannot complete the modification within this time period, please contact me. If you do not wish to modify the manuscript and prefer to submit it to another journal, please notify me of your decision immediately so that the manuscript may be formally withdrawn from consideration by Microbiology Spectrum.

Review AAC tedizolid

The authors report the results of an open label non comparative study which aimed to assess the tolerability and efficacy of long-term treatment with oral Tedizolid for gram-positive prosthetic joint infections (BJIs).

On the basis of their data, the authors conclude that (i) oral tedizolid was well tolerated for prolonged BJIs treatment without significant toxicity, and (ii) clinical failure rate was similar to that of other published BJI investigations.

General comments

There are only limited data about the tolerance and effectiveness of prolonged antibiotic therapy with tedizolid for BJIs. The subject of paper is therefore of potential interest.

My main critics are about (i) the mix of very different types of BJIs about effectiveness and (ii) the very short post-treatment follow-up which both diminish the validity of the conclusions.

Detailed comments

1. BJI is a very vague term that included a lot of different situations associated with distinct outcomes.
2. The study was conducted paper with funding from Merck; I did not see any mention about the authors' COIs
3. Gram-positive pathogens : according to Cochrane = gram-positive (https://community.cochrane.org/book_pdf/295)
4. Page 3 : "Methicillin resistant *S. aureus* (MRSA) are increasingly common » this is not true worldwide (for instance in Western Europe countries)
5. Page 4 : "uncontrolled comorbidities (e.g., diabetes, psychiatric disease)" and "underlying hematologic cytopenias, severe hepatic dysfunction, hypersensitivity to tedizolid or other oxazolidinones, ongoing antibiotic-associated colitis, a diet high in tyramine-containing foods, concurrent use of drugs that may interact with tedizolid (specifically sodium picosulfate or monoamine oxidase inhibitors)," Using all these restriction in the recruitment of the patients decreases the validity about the conclusion on the good profile of tolerability of Tedizolid compared to that of Linezolid as these restrictions correspond to some of the risks factors for Linezolid-related serious adverse events. It also means that the conclusions drawn from the present study only apply to these selected patients.
6. Data about the duration of antibiotics before Tedizolid was started are not presented in the manuscript.
7. Methods : "Finally, we conducted a phone survey 3 months post-tedizolid treatment completion." This is a very short post-treatment follow-up given the nature of the diseases which is likely to have overestimated the remission rate recorded in the study.
8. Page 6 : "For this latter category, the treating physician typically believed continued antibiotics were required for ongoing infection or suspected ongoing infection rather than explicit resolution or failure " is that "suppressive antibiotic therapies "?
9. Regarding Figure 1 : no exclusion case due to tedizolid-resistant strain?
10. What was the microbiology identified in the failure patients (any emergence of linezolid-resistant mutants)?
11. Page 10 : "The lower incidence of thrombocytopenia with tedizolid compared to linezolid » What are the data that support this assertion?
12. Page 12 : "Our study differed from the above observational studies as we used tedizolid for treatment of infection with the intention of clinical cure rather than suppression. Additionally,

unlike the above studies, we systematically surveyed for cytopenias, electrolyte and liver function abnormalities, and neuropathies weekly. » These assertions are incorrect (for instance, tedizolid was not used as suppressive therapy in the Senneville's study which included the biological survey described here).

13. Page 13 : "Finally, while our cure rate was only 35%, ..." "remission" is a more accurate term than "cure" in the settings of BJIs especially in case of chronic infections.
14. The proportion of patients with "ongoing therapy required" is particularly high (51%); the reasons why this prolonged treatment was decided by the physicians need to be described in the manuscript in order to help make a distinction between palliative (stabilization of a failure) and suppressive treatment (maintain of a prolonged remissions).

Reviewer #1 (Comments for the Author):

As prior reviewer #2, I am paying particular attention to the revisions they have made in response to my earlier comments. Overall, they have done a mostly acceptable job addressing these comments. They have brought the study outcomes closer in line with the clinicaltrials.gov site which was my major concern in this unblinded study - specifically that they may not be following their own pre-specified analysis plan. I would still note that the primary outcome measure listed on the clinicaltrials.gov is not reflected in how the manuscript is presented. Pasted here for completeness, the co-primary outcome was defined as:

1) Number of "moderate" or "serious" adverse events as assessed by MedDRA v18.1 [Time Frame: 4-12 Weeks]

2) Number of participants with an outcome of "cure" as defined as no need for further antibiotics beyond the originally planned duration determined by the participant's primary/treating physician. [Time Frame: 4-12 Weeks]

Given this endpoint, I would still expect to see these two measures presented, probably in that order, in the abstract and the manuscript. As in, "x number of solicited moderate or severe adverse events were observed in patients on tedizolid." However the abstract does not provide this. Also that endpoint is not entirely clear since MedDRA is not a tool for grading severity, it's for classifying events into standardized terminology - how was severity assessed?

RESPONSE:

On the clinicaltrials.gov page for this trial (<https://classic.clinicaltrials.gov/ct2/show/NCT03009045>), the primary outcomes are safety and efficacy. Specifically, on this website, it is specified:

“Primary Outcome Measures:

1. Number of "moderate" or "serious" adverse events as assessed by MedDRA v18.1 [Time Frame: 4-12 Weeks]

Study Hypothesis: Tedizolid is well tolerated for prolonged (4-12 weeks) courses of antibiotic therapy for patients with bone and joint infection. Comprehensive Chemistry Panels including liver function tests (CMP) and Complete Blood Count (CBC) Panels will be collected at regular intervals along with standardized surveys to measure adverse events as defined by the Medical Dictionary for Regulatory Activities Terminology (MedDRA) Version 18.1

2. Number of participants with an outcome of "cure" as defined as no need for further antibiotics beyond the originally planned duration determined by the participant's primary/treating physician. [Time Frame: 4-12 Weeks]

Study Hypothesis: Tedizolid is effective for the treatment of bone and joint infection. Specifically, cure will be defined as no need for further antibiotics beyond the originally planned duration (i.e., 6 weeks for non-device associated bone and joint infection or until hardware removal for subjects with implants). Unplanned surgical procedures prompted by inadequate infection control will be categorized as treatment failure. We will also

measure long-term efficacy by performing a phone survey 3 months after completion of antibiotics. Recurrence of signs or symptoms of bone and joint infection will be considered a long-term treatment failure.”

We have modified the manuscript methods to now use language consistent with the clinicaltrials.gov page. Specifically, for safety, the methods now state, “Our primary outcomes were safety and efficacy. Safety was defined as the number of “moderate” and “serious” adverse events using MedDRA 18.1 definitions.” For the results section, we also framed results using the framework outlined in clinicaltrials.gov. Specifically, we now explicitly state, “There were zero moderate or serious adverse events.”

The order of the primary outcomes was already safety followed by efficacy in both the abstract and the methods section. For the sake of brevity in the abstract, we don’t spell out the exact definitions of safety and efficacy, but do explicitly state these outcomes in the methods section of the main manuscript. We think this is sufficient as specifying that the safety outcome was moderate or serious adverse events in the abstract and then reporting there were no moderate or severe outcomes in the methods requires removing other pertinent information. We tried to include the more granular description of primary study outcomes in the abstract, but found we could not without cutting other pertinent information. Importantly, the full information on primary and secondary outcomes is stated in the manuscript, now in a clearer manner that is consistent with wording in on our clinicaltrials.gov page.

The reviewer is correct that MedDRA is not a tool for grading severity. We used the CTCAE system to categorize adverse effects, as is typical in infectious diseases treatment clinical trials. While the CTCAE system was developed for cancer, there is no similar system for antimicrobial treatment trials and the CTCAE system is widely used in NIAID-sponsored clinical trials, such as those supported by the Antibiotic Resistance Leadership Group (ARLG). To clarify the point on severity of adverse events, we have added to the manuscript a reference to the CTCAE system 5.0. Specifically, we state “Severity of adverse events was measured using the CTCAE 5.0 system.”

Minor comments:

I would respectfully submit that despite the claim of careful re-review, they still have not caught or corrected the typo in the abstract results section where it says "...and 5 had (14%) prosthetic joint infection." It should not read "5 had (14%)"

RESPONSE: We apologize for the error. It has been corrected. It now reads, “...and 5 (14%) had...”

Another typo: sentence fragment in the study outcomes section - "Adverse events using MedDRA 18.1 definitions."

RESPONSE: We apologize for the error. It has been corrected. It now reads, “Safety was defined as the number of “moderate” and “serious” adverse events using MedDRA 18.1 definitions.”

Reviewer #2 (Comments for the Author):

The authors report the results of an open label, non-comparative trial of oral tedizolid for bone and joint infections. A total of 37 patients could be evaluated regarding tolerance and efficacy. It was found that prolonged treatment with oral tedizolid was well tolerated and that clinical failure rate was similar to that of other published reports.

General comments

The study is recorded in [Clinicaltrials.gov](https://clinicaltrials.gov)

RESPONSE: Yes, as noted above, the study is recorded in clinicaltrials.gov. The link is: <https://classic.clinicaltrials.gov/ct2/show/NCT03009045>

General comments

The subject is of interest given the paucity of data about the efficacy of tedizolid for the treatment of BJIs.

My main critics are about (i) the mix of very different types of BJIs about effectiveness and (ii) the very short post-treatment follow-up which both reduce the validity of the conclusions.

RESPONSE: We agree that heterogeneity of BJI type can be seen as a limitation. However, BJIs are a difficult disease to study given the heterogeneity of infection types. Some institutions see mostly prosthetic joint infections (PJIs). As a level 1 trauma center, we see a large amount of trauma-associated hardware infections. While the study could have chosen a well defined population (e.g., PJIs), then the results may not be generalizable to other BJIs. Given this study was exploratory (specifically we state it is open label and non-comparative, hence it is not a definitive clinical trial), we think the study design is appropriate and serves as a foundation for more definitive studies with higher quality methods such as blinded, randomized, and specific populations.

In terms of the short follow up, yes, our follow up was only 3 months. Nevertheless, it is longer than most other publications in the literature. We explicitly state this limitation in the manuscript in the limitations section, where we state, "Finally, our follow up was limited and many subjects did not complete the phone survey 3 months after treatment completion. BJI relapse can occur weeks or months after treatment discontinuation, so the true treatment failure rate of tedizolid in our cohort may have been underestimated." Again, we think the short time period is consistent with other prospective studies of BJI treatment and appropriate for an exploratory clinical trial.

Detailed comments

1. *BJI is a very vague term that included a lot of different situations associated with distinct outcomes.*

RESPONSE: We agree that BJI is a broad category. As noted above, we could have chosen a well-defined population (e.g., PJIs), but then the results may not be generalizable to other BJI types. Our study was exploratory in nature and never intended to be a definitive trial such as a randomized controlled double-blind study. One of our primary outcomes was safety, given concerns related to the oxazolidinone class of antibiotics and the paucity of human data

related to prolonged therapy. So this study set out to study in detail the safety in a population of patients with BJIs, since prospective data was lacking.

Of note, the FDA has no definition of a BJI for clinical trials (see <https://www.fda.gov/medical-devices/workshops-conferences-medical-devices/public-workshop-orthopedic-device-related-infections-11132020-11132020>). Thus, by definition BJI is a vague term because its definition is not standardized like other infection types, such as urinary tract infections, intra-abdominal infections, and skin and skin structure infections, all of which have established FDA definitions that guide the development of clinical trials.

2. *The study was conducted paper with funding from Merck*

RESPONSE: Correct. The funding agency and their role (or more specifically lack of role) in study design, study conduct, data analysis, or results interpretation was listed in the acknowledgement section.

3. *Gram-positive pathogens : according to Cochrane = gram-positive*
(https://community.cochrane.org/book_pdf/295)

RESPONSE: We reviewed the linked document and agree that Cochrane uses the above term in the manner that is outlined by the reviewer. We used the format of ASM journals such as *Antimicrobial Agents and Chemotherapy*, which use “Gram-positive” (capitalized and hyphenated). Some recent examples of the capitalized and hyphenated format in ASM journals are: <https://pubmed.ncbi.nlm.nih.gov/37222615/> and <https://pubmed.ncbi.nlm.nih.gov/36946741>. Ultimately, we leave final decision of style to the journal. Until then, we left the formatting as is (and as is consistent with ASM journals). Ultimately, we will let the editorial staff of *Microbiology Spectrum* (an ASM journal) decide how to format this term.

4. *Line 100 : "uncontrolled comorbidities (e.g., diabetes, psychiatric disease)" and "underlying hematologic cytopenias, severe hepatic dysfunction, hypersensitivity to tedizolid or other oxazolidinones, ongoing antibiotic-associated colitis, a diet high in tyramine-containing foods, concurrent use of drugs that may interact with tedizolid (specifically sodium picosulfate or monoamine oxidase inhibitors)," Using all these restriction in the recruitment of the patients decreases the validity about the conclusion on the good profile of tolerability of Tedizolid compared to that of Linezolid as these restrictions correspond to some of the risks factors for Linezolid-related serious adverse events. It also means that the conclusions drawn from the present study only apply to these selected patients.*

RESPONSE: We agree that the definition of the study population limits generalizability, as is the case with all clinical trials given inclusion and exclusion criteria are, by definition, restrictive. To address the reviewer’s valid point, we have added the more common exclusion criteria to the discussion under our limitations paragraph, which now adds, “Third, we excluded patients with hematologic cytopenias, severe hepatic dysfunction, and use of drugs that interact with tedizolid, thus limiting the generalizability of findings to broader populations.”

5. *Data about the duration of antibiotics before Tedizolid was started are not presented in the manuscript.*

RESPONSE: While we did mention prior linezolid therapy in text, we did not summarize prior antibiotic therapy. We agree with the reviewer that this information will help readers understand the patients population that. While information on prior antibiotic therapy outlined in Supplementary Table 1, to be clearer, we added in the results section the following sentence, "Twenty-five (68%) had received prior antibiotic therapy for their BJI; no patient had received prior linezolid therapy. (Supplementary Table 1)"

6. *Line 140: "Finally, we conducted a phone survey 3 months post-tedizolid treatment completion." This is a very short post-treatment follow-up given the nature of the diseases which is likely to have overestimated the remission rate recorded in the study.*

RESPONSE: As noted above, in terms of the short follow up, yes, it is only 3 months.. Senneville et al. claim that they followed their patients treated with tedizolid for BJI for one year, but their manuscript has only one sentence of follow up and do not discuss loss to follow up or other attrition statistics. We explicitly state this limitation in the manuscript in the limitations section, where we state, "Finally, our follow up was limited and many subjects did not complete the phone survey 3 months after treatment completion. BJI relapse can occur weeks or months after treatment discontinuation, so the true treatment failure rate of tedizolid in our cohort may have been underestimated." So we acknowledge this limitation, highlighting it for future investigations that may need longer follow up.

7. *What was the microbiology identified in the failure patients (any emergence of linezolid-resistant mutants)?*

RESPONSE: As outlined in Supplementary Table 1, none of the 4 patients who failed had culturable material or cultures sent to the clinical microbiology lab. Thus, we could not test for linezolid- (or tedizolid-) resistant mutants. We agree with the reviewer that this fact may be of interest to readers. Thus, in the results section, we added a sentence in the manuscript to state, "None of the failures had culturable material, thus we could not test for oxazolidinone-resistant pathogens."

8. *Line 249: "The lower incidence of thrombocytopenia with tedizolid compared to linezolid » What are the clinical data that support this assertion?"*

RESPONSE: We understand the reviewer's concern that this statement is not supported with a reference. Thus, we have added the reference of Bassetti et al. 2019 to that statement. From the abstract of that publication, they state, "Pooled data from ESTABLISH-1 and ESTABLISH-2 indicated a lower frequency of thrombocytopenia in tedizolid-treated than in linezolid-treated patients."

9. *Line 304 : "Finally, while our cure rate was only 35%, ..." "remission" is a more accurate term than "cure" in the settings of BJIs especially in case of chronic infections.*

RESPONSE: We agree with the reviewer that given the nature of BJIs and the risk of recurrence months or even years after initial treatment, "remission" is often a better term than "cure". However, contextually, we discuss cure rate as defined in the clinicaltrials.gov page and the methods section of the manuscript. Thus we think use of the term "cure" in this sentence is

acceptable. We also acknowledge that BJI relapses can occur late in our limitations section in which we state, “Finally, our follow up was limited and many subjects did not complete the phone survey 3 months after treatment completion. BJI relapse can occur weeks or months after treatment discontinuation, so the true treatment failure rate of tedizolid in our cohort may have been underestimated”

10. The proportion of patients with "ongoing therapy required" is particularly high (51%); the reasons why this prolonged treatment was decided by the physicians need to be described in the manuscript in order to help make a distinction between palliative (stabilization of a failure) and suppressive treatment (maintain of a prolonged remissions).

RESPONSE: We agree with the reviewer that the “ongoing therapy required” outcome (51% of study participants) is high. We do think that we explain the reason for this observation in the results section in which we state, “Nineteen (51%) subjects required ongoing therapy at the end of their planned treatment course, typically because of ongoing infection associated with retained hardware that could not be removed after week 12 of tedizolid treatment”. This type of outcome (“ongoing therapy required”) is common in a patient population of trauma-associated hardware infection given “infected” hardware cannot be removed without devastating orthopedic consequences such as loss of limb. Hardware can be removed when bone healing is adequate, a process that may take many months or even years after major orthopedic trauma. We think the details of this description, in conjunction with the narratives of each case in Supplementary Table 1, provide readers with adequate data as to why antibiotic therapy was required beyond our maximum study drug (tedizolid) treatment duration of 12 weeks.

August 7, 2023

Dr. Loren G. Miller
The Lundquist Institute
Infectious Diseases
Harbor-UCLA Medical Center
1000 W. Carson St., Box 466
Torrance, CA 90509

Re: Spectrum01282-23R1 (Safety and Tolerability of Tedizolid as Oral Treatment for Bone and Joint Infections)

Dear Dr. Loren G. Miller:

Your manuscript has been accepted, and I am forwarding it to the ASM Journals Department for publication. You will be notified when your proofs are ready to be viewed.

Sincerely,

Adriana Rosato
Editor, Microbiology Spectrum
